# Perovskite versus ZnCuInS/ZnS Luminescent Nanoparticles in Wavelength-Shifting Layers for Sensor Applications

**DOI:** 10.3390/s21093165

**Published:** 2021-05-02

**Authors:** Aleksandra Sosna-Głębska, Natalia Szczecińska, Maciej Sibiński, Gabriela Wiosna-Sałyga, Bartłomiej Januszewicz

**Affiliations:** 1Department of Semiconductor and Optoelectronic Devices, Lodz University of Technology, 211/215 Wólczanska St., 90-924 Lodz, Poland; natalia.szczecinska@p.lodz.pl (N.S.); maciej.sibinski@p.lodz.pl (M.S.); 2Department of Molecular Physics, Faculty of Chemistry, Lodz University of Technology, 116 Żeromskiego St., 90-924 Lodz, Poland; gabriela.wiosna-salyga@p.lodz.pl; 3Institute of Materials Science and Engineering, Lodz University of Technology, 1/15 Stefanowskiego St., 90-924 Lodz, Poland; bartlomiej.januszewicz@p.lodz.pl

**Keywords:** UV detectors, nanoparticles, down-conversion

## Abstract

In this work, the application of quantum dots is evaluated in order to sensitize the commercially popular Si detectors in the UV range. The wavelength-shifting properties of two types of all-inorganic halide perovskite quantum dots as well as ZnCuInS/ZnS quantum dots are determined in order to assess their potential in the effective enhancement of the sensors’ detection range. In a further part of the study, the wavelength-shifting layers are formed by embedding the quantum dots in two kinds of polymers: PMMA or Cyclic Olefin Polymer. The performance of the layers is evaluated by transmission and PLE measurement. Incorporating the nanoparticles seemingly increases the transmittance in the UV range by several percent. The observed phenomenon is proportional to the quantum dots to polymer concentration, which indicates the successful conversion action of the luminescent agents.

## 1. Introduction

Ultraviolet (UV) photon detection is currently becoming increasingly important in many fields of application. These are, among others, UV index monitoring, UV curing, UV warning and explosion detection systems, UV photography, UV–vis spectroscopy and, last but not least, nuclear physics, in which the energy of scintillated particles corresponds to the UV range. The semiconductor-based UV–vis sensor is traditionally configured with Si material, although its sensitivity in UV is quite poor—typically, for an unbiased detector, it is around 0.1 A/W, which is less than 20% of the maximum responsivity value [1,2,3]. This can be observed in Figure 1, showing the current responsivity of two popular types of Hamamatsu silicon detectors.

To detect UV light, photodetectors based on wide-bandgap materials such as GaN, SiC, ZnO [4] can also be used. They are visible-blind UV detectors, which is often an advantage, although in some applications, visible range detection is also required.

Nevertheless, the detectors which are sensitive in a continuous broad spectral range of UV to visible, which can operate without bias, are lightweight and fairly cheap and are in high demand. A solution to this problem could be the application of the so-called wavelength-shifting materials on top of the commercially available and low-cost silicon detectors. These materials deposited on the top of such a detector would absorb the UV radiation and convert it into the visible wavelength range to which the detector underneath is highly sensitive, effectively extending the useful wavelength detection range. Organic chemical wavelength shifters, such as Tetraphenyl Butadiene (TPB), have been proposed for such applications [5]. Unfortunately, they possess serious drawbacks, such as instability in noble environments, which is often required for nuclear physics experiments, photo-degradation under UV light and limited Stokes shift, which causes the re-absorption of the generated light [6].

The materials for this kind of wavelength-shifting application should ideally possess features such as a broad excitation range for UV light (200–450 nm) and a narrow emission peak matched to the highest conversion range of the silicon photodetectors, which is 500–600 nm. At the same time, they should have high photoluminescence quantum yield, but no parasitic absorption in the visible range. Additionally, in order not to distort the measurement when deposited on the detector, they should have a photoluminescence lifetime that is as short as possible and be chemically stable.

The materials which could be considered for this application can be divided into the following subgroups:Metal oxides such as zinc oxide (ZnO) and titanium dioxide (TiO_2_) have an energy bandgap corresponding to the UV range absorption (3.37 eV for ZnO, 3.2 eV for TiO_2_ anatase and 3.0eV for TiO_2_ rutile [7]). Thanks to the defect-related states, the emission in the visible range can be observed [8,9,10]. While doping them with different metals, an increase in the PLQY can be observed. For example, for lightly S-doped ZnO powder, an efficiency of 55% has been reported, increasing up to 75% at higher doping concentration [11]. Another approach is taking advantage of miniaturization. The defect states of metal oxide nanoparticles can be controlled by the synthesis method. Recently, ZnO/polyacrylic acid (PAA) nanohybrids in silica coating of PLQY of 64% have been reported [12].Metal oxides doped with rare earth ions, in which the origin of photoluminescence is the intraband 4f-4f transitions as well as 4f-5d interband transitions [13,14]. They are the state-of-the-art phosphors researched for application in white LEDs and can achieve PLQY close to 80–90% with satisfactory excitation–emission characteristics [15]. Among this group, the best known persistent luminescent phosphors can be found with afterglow of several hours [16]. Their application in luminescent solar concentrators for photovoltaics has been widely researched [17,18,19] also by the authors [20]. For enhancing the real-time photoresponse of photodetectors, the types with the shortest possible photoluminescence lifetime should be considered.State-of-the-art quantum dots of different kinds. Among all types of nanostructures, quantum dots possess the highest density of states and therefore can exhibit high luminescence efficiency close to 100% [21]. Especially desirable for the considered application, they also possess very narrow emission spectra, which can be tailored by their size. Cd-based QDs are probably the most researched luminescent nanomaterial thanks to their ease of synthesis and suitable optical properties [17]. Another widely researched wavelength-shifting material is Pb-based QDs. Moreover, all-inorganic lead halide perovskite QDs (IPSK QDs) have emerged as attractive candidates for wavelength-shifting agents due to their controllable and high-intensity PL, high optical absorption coefficient [10], high chemical stability and single-component ultra-fast decay time of the order of ns at room temperature [22]. Their successful application as wavelength-shifting agents has been proven, among others, in photovoltaics [23,24] and high-energy physics [25]. Despite the excellent optical properties of Pb- and Cd-based materials, their application on a large scale is unlikely due to the toxicity of lead and cadmium. The best available heavy-metal free quantum dots are based on ZnS, ZnSe, InP, Si, CuInS2 and, recently popular due to their high abundance, low toxicity and large possibility of tailoring the bandgap, graphene and carbon QDs. For ZnSe QDs, a PL efficiency equal to 70% has been reported [26], and for carbon QDs, the achieved efficiency is as high as 94% [27]. ZnCuInS/ZnS core-shell QDs, which are considered in the frame of this article, are hydrophobic quantum dots [28] exhibiting tunable emission in the visible spectral window from around 500 to 620 nm [29] and PLQY up to 81% [30].

However, the construction of complete converting layers and, furthermore, functional devices based on these materials is a challenging and unsolved problem. Apart from choosing a suitable wavelength-shifting agent, another task is to evenly distribute and attach it to the detector surface. To ensure the mechanical and chemical stability of the wavelength-shifting layer, a transparent polymer can be used as a matrix in which the luminescent nanoparticles can be embedded.

In the frame of this work, the authors propose and test technological steps for the practical formation of the wavelength-shifting layers consisting of luminescent nanoparticles immobilized in polymer. The experiments involve three kinds of light-converting agents: two types of Pb-based perovskite QDs and non-toxic ZnCuInS/ZnS QDs. Two polymer kinds are tested: PMMA, which is omnipresent in our everyday lives due to easy handling, processing and low cost; and Zeonex^®^, a Cyclic Olefin Polymer (COP), which possesses higher transparency than PMMA in the UV range [31]. The layers are examined for their application as light-sensing wavelength-shifting layers suitable for the enhancement of the effective spectral range of silicon photodetectors. By this method, the authors hope to achieve a relatively inexpensive and scalable technology for sensor array modification.

## 2. Materials and Methods

The polymers used for the base layers were: PMMA of avg. mol wt. ~350,000 g/mol purchased from Sigma Aldrich and Cyclic Olefin Polymer ZEONEX^®^ 480 purchased from Zeon Europe GmbH. Toluene pure p.a., which was used as a solvent, was purchased from Chempur. All quantum dots were purchased from PlasmaChem GmbH: 2 kinds of perovskite quantum dots with emission maximum 510 ± 15 nm (referred to later as PSK 515 QDs) and 550 ± 15 nm (referred to later as PSK 525 QDs) as well as Zn-Cu-In-S/ZnS quantum dots with emission maximum 590 ± 15 nm. Their emission characteristics were chosen in order to spectrally match the maximum response of the commonly used silicon detectors [2,3].

The preparation procedure of a thin polymer layer with quantum dots is schematically illustrated in the Figure 2. A polymer base composed of polymer granulate and the solvent (toluene) is stirred for 24 h with a magnetic stirrer. After this time, a certain volume of quantum dots dissolved in toluene is added to the polymer base solution. To keep the ratio of polymer to toluene constant in all the samples, pure toluene is added at this stage as well. Such mixtures are subsequently stirred in an ambient atmosphere for at least 2 h before deposition with the spin-coating method and then pre-heated on a hot-plate at 50 °C. Every mixture is deposited on two kinds of substrates: 2 × 2 cm JGS2 1 mm thick quartz squares purchased from Ameko-tech and 1.5 × 1.5 cm pieces of silicon n-type wafer in orientation {100} for better characterization and simulation of work conditions.

For every series of samples, the polymer base marker sample is deposited—in this sample, there are no quantum dots. The polymer bases weighted at stage 1 are 10% wt. Zeonex and 16% wt. PMMA. At stage 3, the base solutions are diluted, so that the final polymer to toluene proportions are 5% for Zeonex and 8% for PMMA. The bases’ compositions were chosen after the optimization procedure, in which a series of samples differing in proportions were characterized optically and mechanically. The optical transmittances of the chosen bases in the full operation range are shown in Figure 3.

The transmittance measurements were performed with a Filmetrics aRTie-UV thin-film analyzer with a deuterium–tungsten source, model LSDT2. The characterization of the excitation–emission characteristics was performed using a FLS980 Edinburgh Instruments (Livingston, UK) fluorescence spectrometer with an excitation source constituted by an ozone-free 450 W Xenon Arc Lamp and a photosensitive element R-928 photomultiplier detector by Hamamatsu. The SEM photographs were taken with a JEOL JSM-6610LV scanning electron microscope equipped with an OXFORD X-MAX 80 detector, enabling EDX analysis.

## 3. Results

### 3.1. Characterization of the Wavelength-Shifting Agents

The PL spectra were measured at room temperature in a quartz cuvette using diluted solutions of QDs in toluene. They are depicted in the form of excitation–emission maps in Figure 4.

The normalized emission and excitation spectra of three kinds of quantum dots are compiled in Figure 5.

A drop of each solution of quantum dots in toluene was spin-coated on a silicon substrate in order to observe the morphology as well as to establish the chemical composition. The quantum dots were grouped in bigger agglomerates, photos of which are shown in Figure 6. As the chemical composition of perovskites is unknown, EDX analysis was used to determine the constituting elements.

### 3.2. Wavelength-Shifting Layers

The list of samples, which have been prepared according to the procedure presented in Figure 2, is presented in Appendix A, Appendix A.

The exemplary SEM photographs of two samples are shown in Figure 7**.**

The luminescent material was grouped in bigger agglomerates with diameters of a few tens of µm. The thickness measurements revealed that Zeonex layers were thicker, at around 2 µm, whereas PMMA layers were thinner, at 800 nm, and in this case, the polymer did not cover the agglomerates at the top.

The qualitative measure to assess the wavelength-shifting effect in the deposited layer is the transmittance measurement. Its outcome cannot be treated as the exact transmittance of the sample due to the fact that the result is distorted by the luminescent properties of the sample. This is why the measured transmittance is “enhanced” in the UV range, which means in fact that the UV light is absorbed and converted to wavelengths at which the detector in the set-up is more sensitive. This is illustrated in Figure 8.

In Figure 9, Figure 10, Figure 11, Figure 12, Figure 13 and Figure 14, the transmittance measured as explained above is presented—the results are an average of four measurements. Subfigures (a) show the absolute transmittance for every sample in the series with two reference curves: the transmittance of the quartz substrate and the quartz substrate coated with the polymer base without any luminescent materials. Different percentage values correspond to the volume ratio of the QDs in toluene solution to the polymer base in toluene solution. The exact composition of the samples can be found in Appendix A, Appendix A. Subfigures (b) present the sample to the polymer base transmittances ratio. The wavelength ranges and the samples where the ratio exceeds 1 are particularly interesting, as this is where the “positive” influence of light-converting agents is visible.

Based on the transmittance measurement results, we can distinguish three particularly interesting regions:UV-C range (190–230 nm), where a strong enhancement can be observed (around 10%), especially series PSK515 in PMMA (Figure 9) and PSK525 in Zeonex (Figure 12).240 nm, at which a local minimum can be observed in some samples.UV-B and UV-A range, where a stable and slight enhancement (up to 2–3%) of the transmittance is observed. The wavelength range 270–500 nm corresponds to the excitation region of the measured quantum dots.

The dependence of the transmission at one characteristic wavelength of each region (200 nm, 240 nm and 300 nm) is plotted versus the polymer to QDs ratio in Figure 15**.**

In all series of samples, the greatest dispersion of points can be observed at 200 nm. It is usually equal to 10%, apart from PSK515 in PMMA (Figure 15a), where it reaches even 30–40%. However, the UV-C wavelength range is also very sensitive to the polymer thickness, which could be the reason for the unsystematic results in this range. The transmittances at 240 nm and 300 nm are more concentration-dependent, especially in the case of perovskite in Zeonex QDs. To analyze the relation between the amount of luminescent material and its influence on the transmittance of the sample more thoroughly, the transmittance in the excitation range of the quantum dots, i.e., 270–500 nm, has been integrated and plotted versus the polymer to luminescent material mass ratio. The dependence has been fitted with a third-order polynomial curve in the case of PMMA (Figure 16), and a linear fit in the case of Zeonex (Figure 17).

It can be noticed that there is no strong dependence in the case of QDs dispersed in PMMA, although the enhancement is visible for some samples. For PSK515 and PSK525 in the Zeonex series, the influence of the luminescent material concentration on the integrated transmission in the excitation range of the perovskites can be quite accurately modeled until the mid-range of the Zeonex to QD mass ratio with the linear fit. The Zeonex to perovskite mass ratio until which the integrated transmittance is higher than the transmittance of the base layer is equal to around 3000.

Among all the analyzed samples, it was the sample of PSK515 in Zeonex (120%) which exhibited the highest increase in integrated transmittance. Therefore, it was subjected to further photoluminescence measurements, the results of which are shown in Figure 18.

From Figure 18a, it can be seen that apart from the strong peak at 510 nm attributed to the perovskite quantum dots, there is also much weaker emission in the 400–460 nm range, which can be attributed to the Zeonex base layer. The excitation spectra, depicted in Figure 18b, show that the perovskite quantum dots are excited in the whole UV range, but more efficiently in the UV-B than UV-A region, as planned, whereas the Zeonex base is most efficiently excited by the 370 nm wavelength.

The same sample was subjected to additional optical experiments. Transmittance measurement with the UV filter eliminating all the light below 320 nm was performed.

At first, the filter was placed below the sample, blocking all the UV-B light transmitted through. Then, the filter was placed above the sample so that the UV-B light did not induce the wavelength-shifting effect. This is schematically illustrated in Figure 19a. The signal ratio with the UV filter below/above the sample shows that there is a difference in the signal of around 1% for the PSK515 in the Zeonex sample, whereas the Zeonex base layer shows no difference in the UV filter position.

## 4. Discussion

The observed “increase” in the transmittance of the sample is in fact a result of the anticipated wavelength-shifting effect. However, the obtained results depend on the spectral responsivity of the detector integrated in the set-up used for the transmittance measurement, which remains unknown. Although, in some samples, an increase in transmittance when compared to the polymer base layer occurred, all of them were transmitting less in the whole spectral range when compared to the quartz substrate. Therefore, when implemented on a real device, they would deteriorate its performance rather than enhancing it. However, to demonstrate the potential of the wavelength-shifting layer, we evaluated theoretically the influence of depositing the three considered luminescent nanoparticles without a polymer matrix on the surfaces of two popular silicon Hamamatsu detectors. Si 227-16 BQ is a traditional silicon photodiode, composed of a single p-n junction. MPPC 13360-1350 CS is a Multi Pixel Photon Counter (MPPC), a type of silicon photomultiplier, used for photon counting and other ultra-low-light applications. It is composed of a high-density matrix of avalanche photodiodes, which, when operated in Geiger mode, generate the high internal gain that enables single photon detection. Their responsivity is presented in Figure 1.

The detectors’ modified responsivity $R_{i_{m}}\left(\lambda_{ex}\right)$ is modeled using the following formula:(1)$$R_{i_{m}}\left(\lambda_{ex}\right)=T\left(\lambda_{ex}\right)\cdot R_{i}\left(\lambda_{ex}\right)+PLQY\cdot A\left(\lambda_{ex}\right)\cdot \int_{\lambda_{em\_end}}^{\lambda_{em\_0}}Em_{\lambda_{ex}}(\lambda_{em})\cdot R_{i}\left(\lambda_{em}\right),$$
where $T\left(\lambda_{ex}\right)$ is transmittance, $R_{i}\left(\lambda_{ex}\right)$ is the original responsivity, PLQY is the photoluminescence quantum yield, $A\left(\lambda_{ex}\right)$ is the absorbance, $\int_{\lambda_{em\_end}}^{\lambda_{em\_0}}Em_{\lambda_{ex}}(\lambda_{em})\cdot R_{i}\left(\lambda_{em}\right)$ is the integral of the emission spectrum multiplied by the responsivity of the detector.

For the simplified calculations, the following assumptions were made:
Absorbance $A\left(\lambda_{ex}\right)$ of the luminescent material is approximated by the excitation spectrum, as shown in Figure 5. The maximum absorbance is set to 1. This approximation is based on the absorption measurement, which demonstrated that the absorption and excitation spectra correlated with each other.Transmittance $T\left(\lambda_{ex}\right)$ is equal to $1-A\left(\lambda_{ex}\right)$. For the wavelengths where there is no absorption, the transmittance is set to 1. This assumption is used due to the fact that the transmittance measured for coatings is in fact modified by the wavelength-shifting effect. Reflectance and scattering are excluded from this model, as it is based only on the experimental measurements of the luminescent materials and not on the deposited coatings.The shape of the emission spectrum is invariant with the excitation wavelength; therefore, the integral $\int_{\lambda_{em\_end}}^{\lambda_{em\_0}}Em_{\lambda_{ex}}(\lambda_{em})\cdot R_{i}\left(\lambda_{em}\right)$ is constant for a particular luminescent material and a particular detector type. This is supported by the excitation–emission measurements.The PLQY is set to 10% in order to compensate all the “optimistic” assumptions described in the previous points as well as to “smooth” the curves presented in Figure 18.

Although the results presented in Figure 20 are based on so many unknowns that they cannot be treated quantitatively, one can see that the potential of responsivity gain in the UV range is in fact very significant.

The positive influence of luminescent nanoparticles on the photoresponse has been already proven experimentally. CdTe QDs deposited directly on the MPPC detectors’ surface caused a fivefold increase in the signal in the 200–230 nm range and the application of transparent tape coated with different kinds of ZnS NPs increased the detectors’ response in the range of 210–300 nm, attaining a fourfold increase at 225 nm [6]. Perovskite CsPbBr3 nanoparticles drop-casted on UV-quartz and placed together with a Si-based photodetector in an integrating sphere caused a nearly threefold improvement in responsivity at 270 nm [32]. However, forming a chemically stable and mechanically resistant wavelength-shifting layer with luminescent nanoparticles embedded in a matrix is still a challenge. There have been some successful attempts in this field, such as the purple-emissive carbon quantum dots dispersed in a PVA layer deposited on a Si photodetector, which relatively increased its photoresponse at 290 nm by 203.8% [33].

## 5. Conclusions

In the present work, the authors evaluated the possibility of the practical application of halide perovskite quantum dots as well as ZnCuInS/ZnS quantum dots for the extension of widely used Si detectors’ sensitivity range. At the initial stage, the optimum material composition was verified, and the possible layer structure, processing and deposition technique were proposed. Based on two types of polymer carrier, a series of layers were deposited on the tested quartz substrates. Through in-depth analysis of the optical properties of these samples, the desired effect was confirmed and specific conditions of layer manufacturing process were determined. Additionally, interesting dependences of the influence of the luminescent material concentration in Zeonex on the integrated transmission in the excitation range of the perovskites were observed for the first time. Based on the positive effect of QD application on UV light conversion, the authors conducted their analytical evaluation of the dependence of the spectral responsivity of the selected detectors (Si 227-16 BQ and MPPC 13360-1350 CS) on the investigated layers’ composition. In this case, significant progress in the UV range was ultimately confirmed.

## Figures and Tables

**Figure 1 sensors-21-03165-f001:**
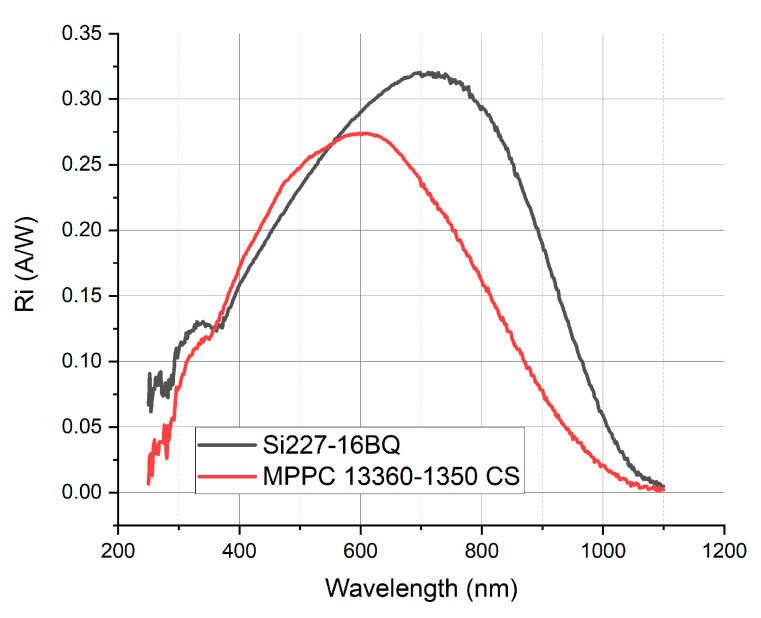
Current responsivity of two kinds of unbiased silicon Hamamatsu detectors measured by the authors with the PVE300 Bentham EQE set-up.

**Figure 2 sensors-21-03165-f002:**
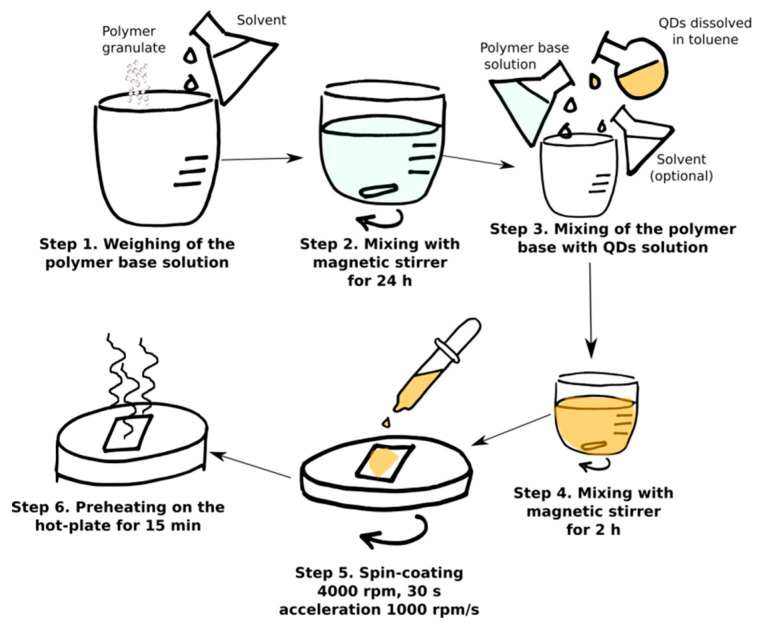
Schematical representation of the preparation of luminescent layers with quantum dots dissolved in toluene.

**Figure 3 sensors-21-03165-f003:**
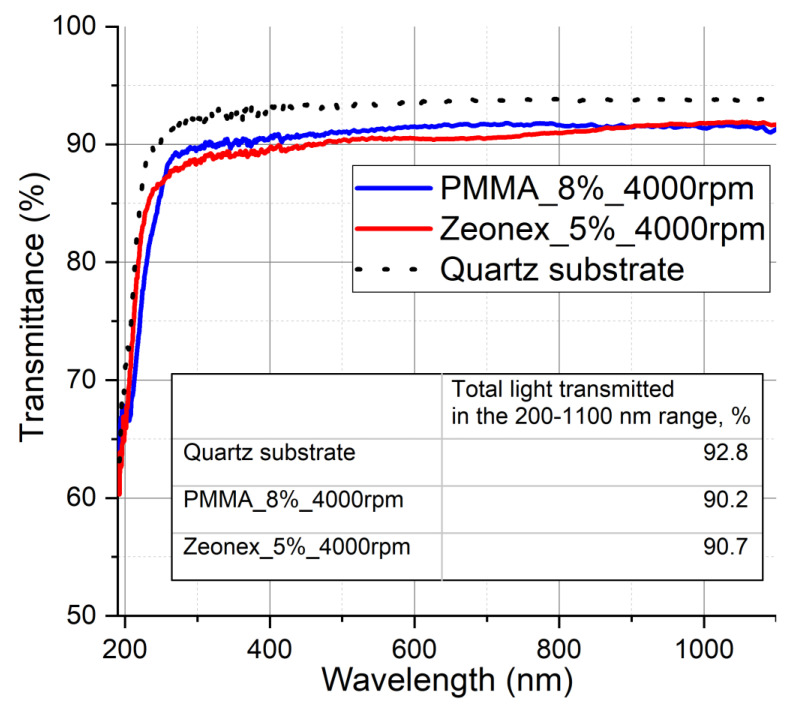
Transmittance characteristics of the base layers without quantum dots.

**Figure 4 sensors-21-03165-f004:**
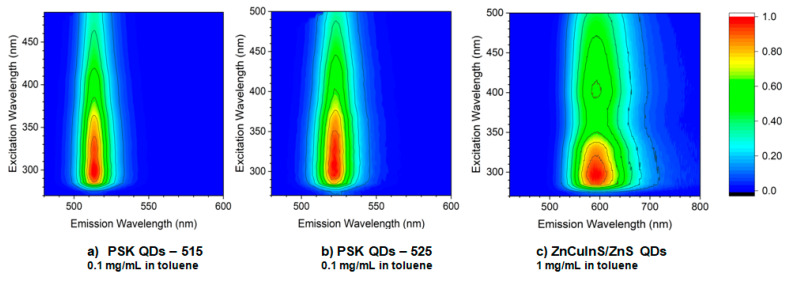
Excitation–emission maps for (**a**) perovskite quantum dots with maximum emission at 515 nm dissolved in toluene at concentration 0.1 mg/mL, (**b**) perovskite quantum dots with maximum emission at 525 nm dissolved in toluene at concentration 0.1 mg/mL, (**c**) ZnCuInS/ZnS quantum dots with maximum emission at 598 nm dissolved in toluene at concentration 1 mg/mL.

**Figure 5 sensors-21-03165-f005:**
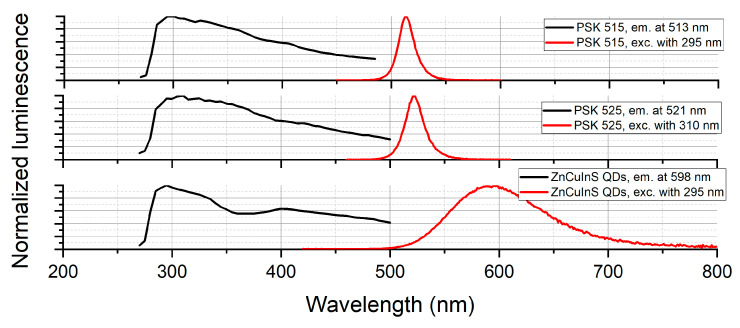
Excitation and emission characteristics for 3 considered materials at most optimal excitation and emission wavelengths.

**Figure 6 sensors-21-03165-f006:**
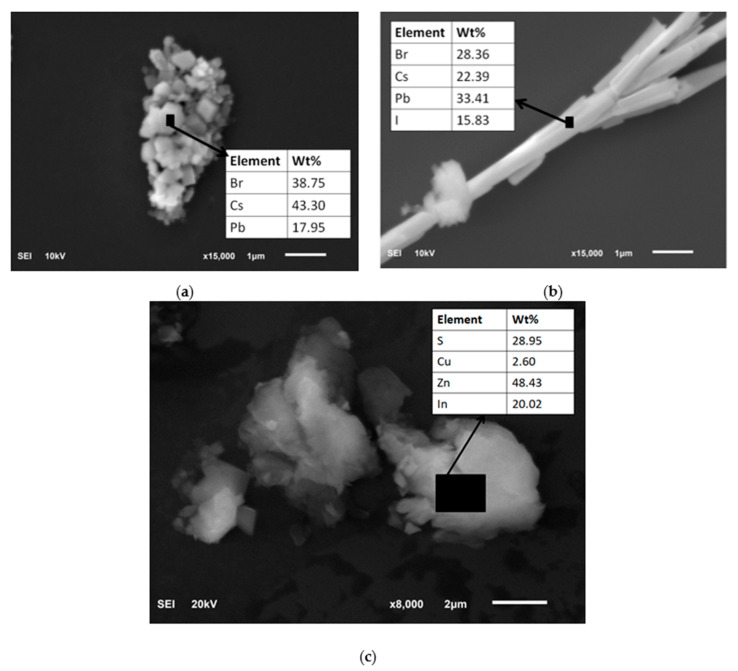
SEM photographs with composition analysis of (**a**) PSK 515 QDs, (**b**) PSK 525 QDs and (**c**) ZnCuInS/ZnS QDs listed in Table 1.

**Figure 7 sensors-21-03165-f007:**
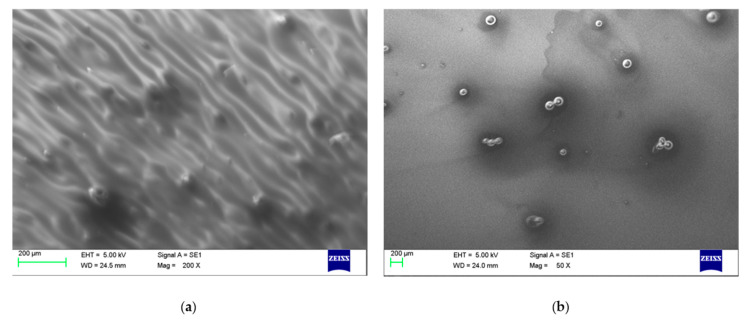
SEM photographs of (**a**) PSK515_Zeonex_50% sample and (**b**) PSK525_PMMA_100% sample.

**Figure 8 sensors-21-03165-f008:**
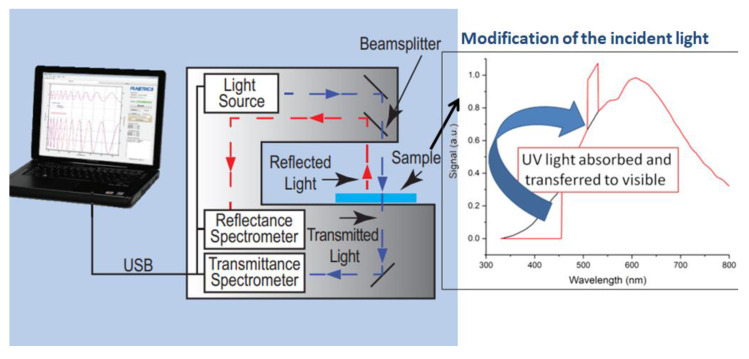
Filmetrics aRTie-UV thin-film analyzer (producer’s scheme) with the wavelength-shifting sample modifying the incident spectrum.

**Figure 9 sensors-21-03165-f009:**
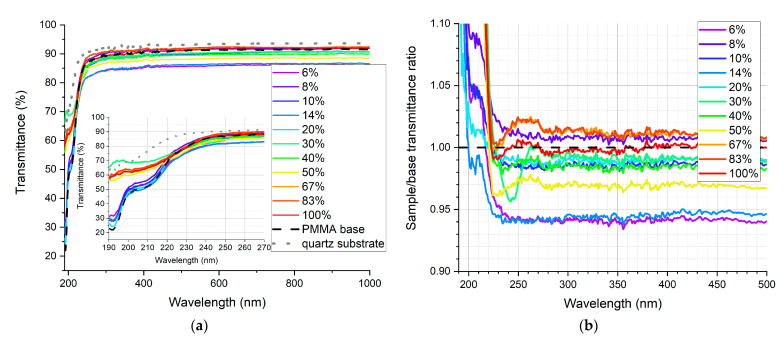
(**a**) Absolute transmittance of each sample in the PSK515 in PMMA series with 190–270 nm range in the inner graph, (**b**) sample to base transmittance ratio.

**Figure 10 sensors-21-03165-f010:**
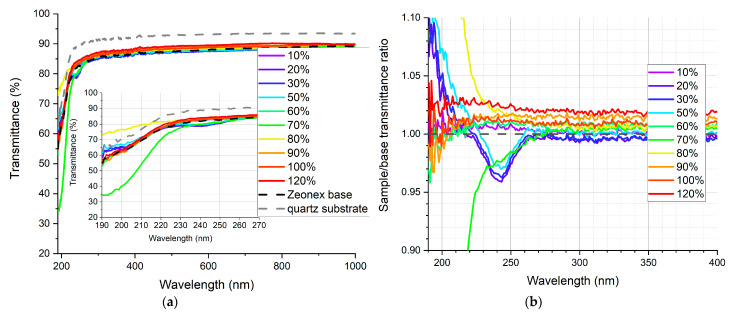
(**a**) Absolute transmittance of each sample in the PSK515 in Zeonex series with 190–270 nm range in the inner graph, (**b**) sample to base transmittance ratio.

**Figure 11 sensors-21-03165-f011:**
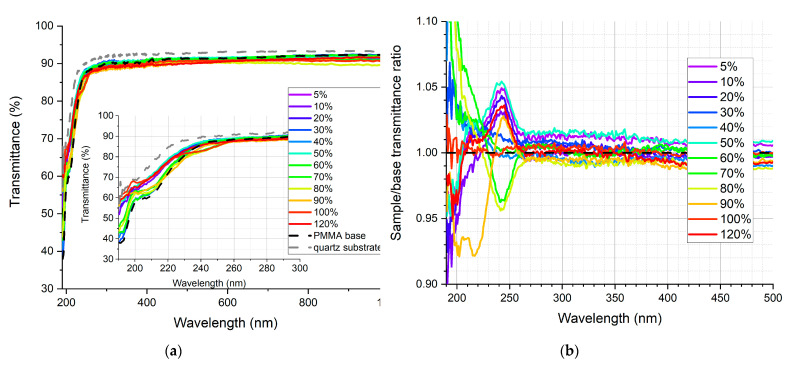
(**a**) Absolute transmittance of each sample in the PSK525 in PMMA series with 190–290 nm range in the inner graph, (**b**) sample to base transmittance ratio.

**Figure 12 sensors-21-03165-f012:**
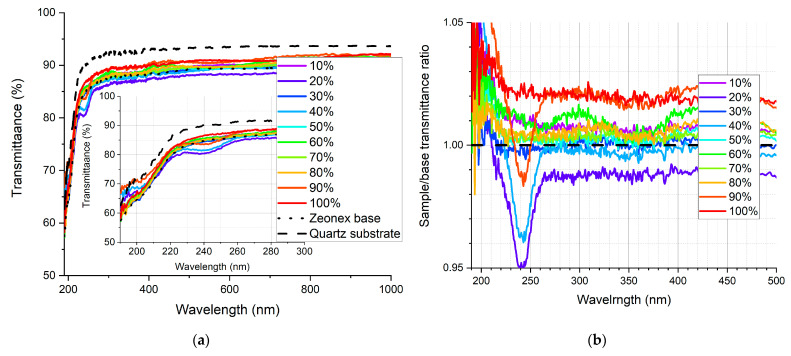
(**a**) Absolute transmittance of each sample in the PSK525 in Zeonex series with 190–300 nm range in the inner graph, (**b**) sample to base transmittance ratio.

**Figure 13 sensors-21-03165-f013:**
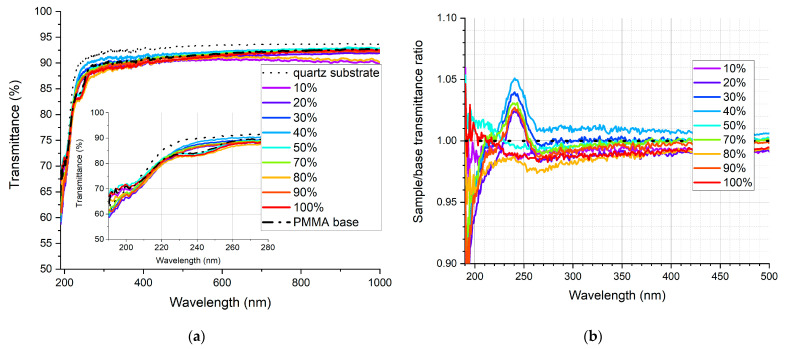
(**a**) Absolute transmittance of each sample in the ZnCuInS/ZnS QDs in PMMA series with 190–270 nm range in the inner graph, (**b**) sample to base transmittance ratio.

**Figure 14 sensors-21-03165-f014:**
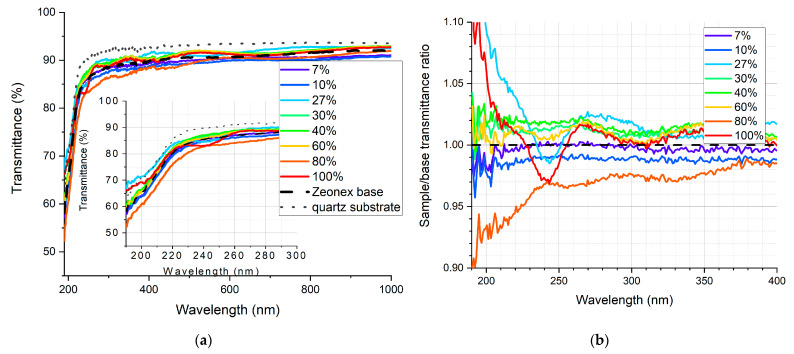
(**a**) Absolute transmittance of each sample in the ZnCuInS/ZnS QDs in Zeonex series with 190–300 nm range in the inner graph, (**b**) sample to base transmittance ratio.

**Figure 15 sensors-21-03165-f015:**
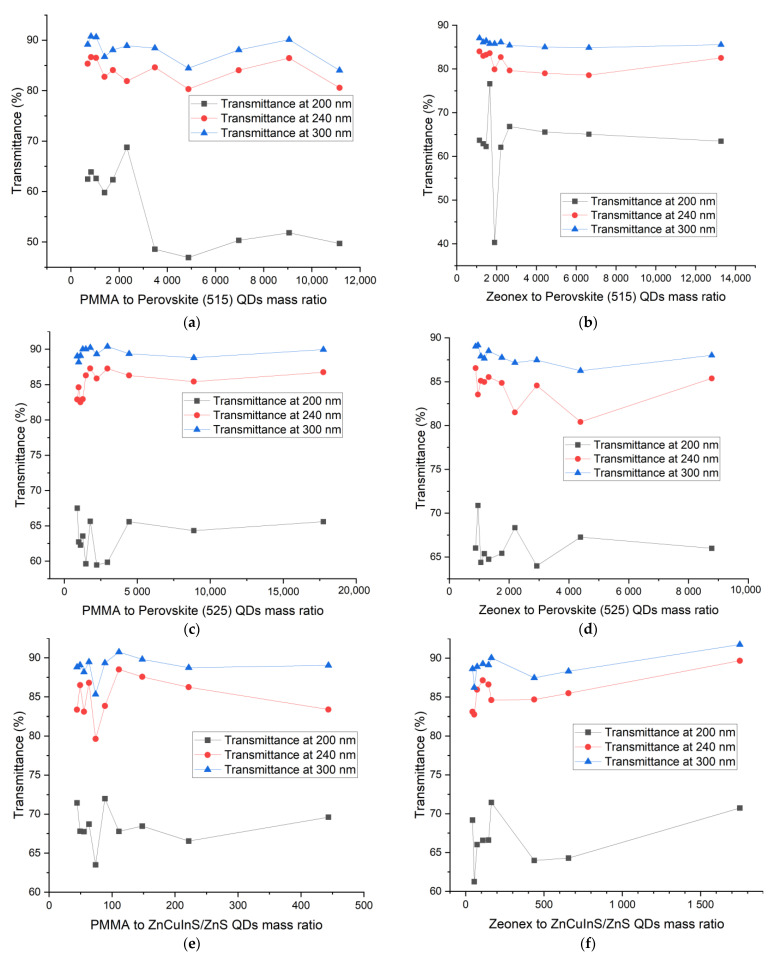
Transmittance at 3 characteristic wavelengths (200, 240 and 300 nm) versus the polymer to quantum dots ratio for (**a**) PSK515 in PMMA series, (**b**) PSK515 in Zeonex series, (**c**) PSK525 in PMMA series, (**d**) PSK525 in Zeonex series, (**e**) ZnCuInS/ZnS in PMMA series, (**f**) ZnCuInS/ZnS in Zeonex series.

**Figure 16 sensors-21-03165-f016:**
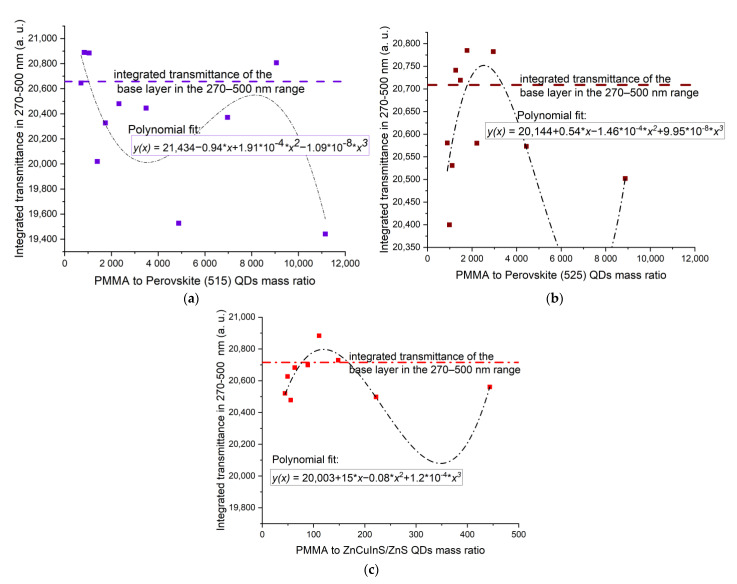
Integrated transmittance in 270–500 nm range versus PMMA to QD mass ratio for (**a**) PSK515 in PMMA series, (**b**) PSK525 in PMMA series, (**c**) ZnCuInS/ZnS in PMMA series.

**Figure 17 sensors-21-03165-f017:**
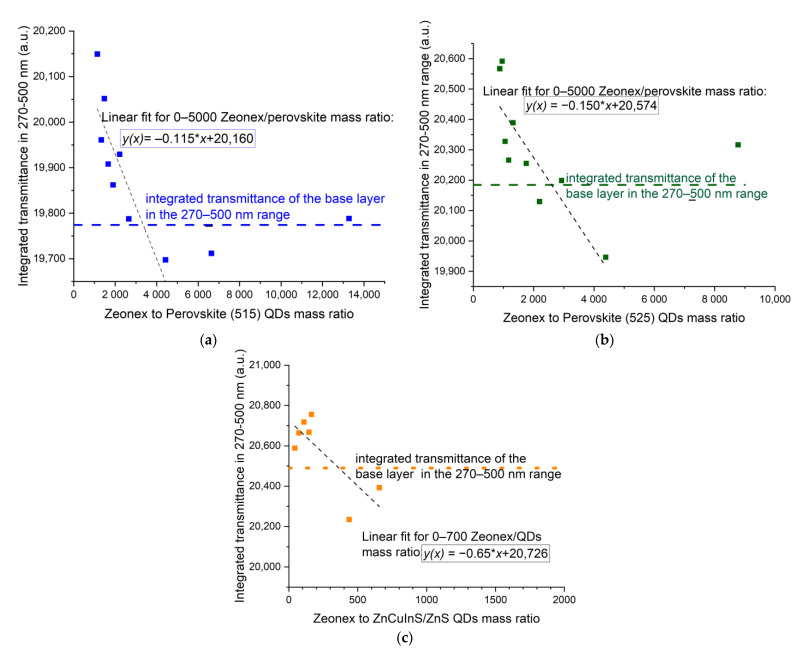
Integrated transmittance in 270–500 nm range versus Zeonex to QDs mass ratio for (**a**) PSK515 in Zeonex series, (**b**) PSK525 in Zeonex series, (**c**) ZnCuInS/ZnS in Zeonex series.

**Figure 18 sensors-21-03165-f018:**
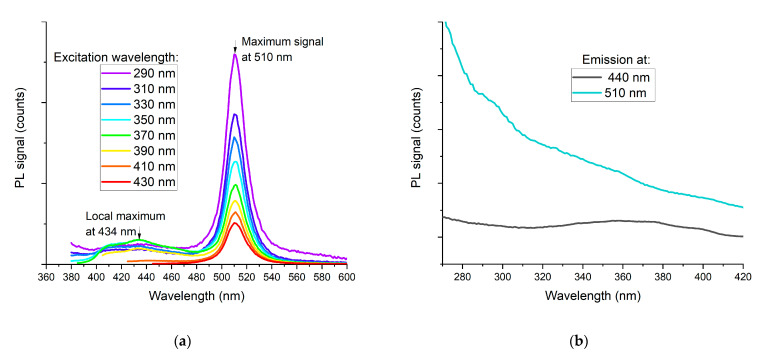
(**a**) Emission spectra and (**b**) excitation spectra at two local maxima of PL spectrum for sample PSK515 in Zeonex—120%.

**Figure 19 sensors-21-03165-f019:**
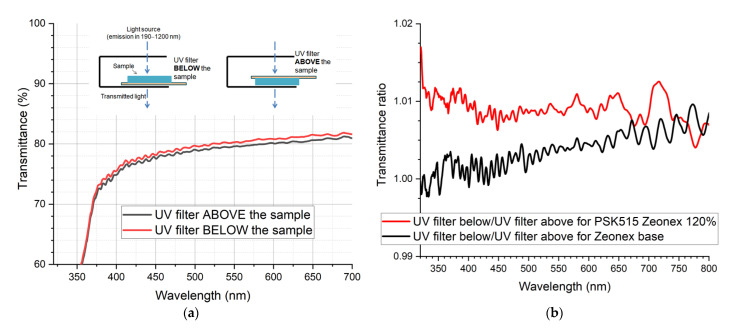
(**a**) Absolute transmittance with UV filter placed above and below the PSK515 in Zeonex 120% sample and (**b**) the transmittance ratio of these two measurements.

**Figure 20 sensors-21-03165-f020:**
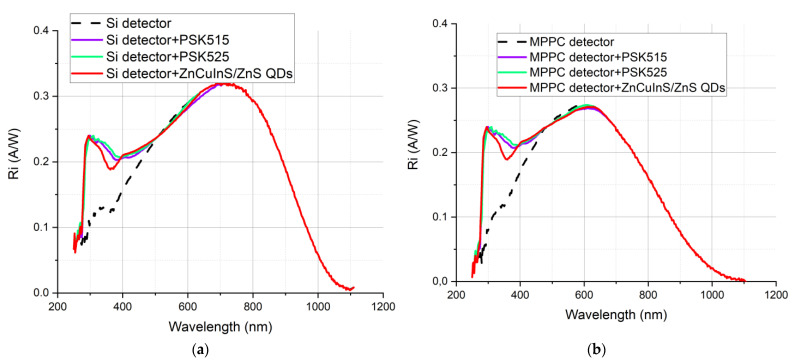
Simulations of possible responsivity enhancement of 3 types of considered luminescent materials deposited on (**a**) Si 227-16 BQ and (**b**) MPPC 13360-1350 CS detector.

**Table 1 sensors-21-03165-t001:** Characteristics of considered wavelength-shifting materials.

Short Name	Composition	Average Particle Size (Producer Data)	Photoluminescence Quantum Yield (Producer Data)	Excitation Range	Maximum Emission	FWHM of Emission Peak
PSK QDs 515	CsPbBr_3_	~10 nm	~90%	270–500 nm	513 nm	16.5 nm
PSK QDs 525	CsPb(I/Br)_3_				521 nm	21.5 nm
ZnCuInS QDs	Zn-Cu-In-S/ZnS	4–5 nm	15–30%		598 nm	98 nm

## Data Availability

Not applicable.

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
