# Peer review of "Perovskite versus ZnCuInS/ZnS Luminescent Nanoparticles in Wavelength-Shifting Layers for Sensor Applications"

_sensors, 2021, doi:10.3390/s21093165_

Round 1

Reviewer 1 Report

This paper investigates a interesting topic of wavelength conversion using QDs to boost the UV detection efficiency of Si photodetectors. While the topic is interesting and the introduction and experimental procedure were well presented, the experimental data and discussions are confusing and do not provide notable technical advance in this field. The readers would be very interested to known the performance comparison of these QDs yet this is not clearly presented. The reviewer has two major concerns:

(1) While PL and PLE analyses are performed in QD solutions, there is no data showing the PLE or PL of spin-coated thin films. The latter would be much more relevant to evaluate the efficiency of wavelength conversion from UV to visible regime. These data must be presented, including electron microscopy images to verify the uniform distribution of the QDs, the thickness of the films, as well as the PL intensity/efficiency comparison among different types of QDs.

(2) The paper mostly focused on the "enhanced" UV transmission after incorporating QDs, which is also very confusing. To implement wavelength conversion efficiently, the QD should absorb UV photons and emit in visible regime, which means the UV transmittance should be decreased rather than increased. This leads to the question as to whether the coating thickness are indeed identical between the reference sample without QDs and those with QDs, as well as the uniformity of QDs in the coatings. Without such verification, the comparison could be meaningless.

(3) The enhanced responsivity on Si photodetector is purely based on modeling. Very little directly measured experimental data was incorporated. This is not very convincing. At least the authors should use experimentally measured transmittance spectra and the measured PLE efficiency for such modeling. Reflectance should also be measured since Absorptance A=1-T-R. This is important for thin coatings. 

Based on these considerations, the reviewer suggests major revisions before further consideration.

Reviewer 2 Report

The authors have synthesized and evaluated halide perovskite quantum dots as well as ZnCuInS/ZnS quantum dots for extension of Si detectors sensitivity range. Some basic material characterizations have been performed and the authors carried out analytical evaluation of dependence of the spectral responsivity of the selected detectors (Si 227-16 BQ and MPPC 13360-1350 CS) on investigated layers composition. However the authors need to address the following questions to make the paper more clear to the readers.

  • Figure 6, what is the uniformity of the QDs spin-coated on quartz/ silicon substrate;
  • Figure 7-12, the figures are very blur and the inset graphs are unclear. In addition, can the authors be more specific on the subfigure (a): what is the 6%-100% legend stands for?
  • Overall the sample to base transmittance ratio for different QDs in different polymers have different curves (figure 7-12 (b)). However there is some enhancement as well as some dip in the sample to base transmittance ratio curve. There are no rules on the curve (e.g. the same wavelength for the dip/lowest sample to base ratio and trend on the luminescent material concentration). Can the authors provide a more analysis on the reason other than shown the unsystematic results?
  • The authors have simulated of possible responsivity enhancement of 3 types of considered luminescent materials deposited on selected Si photodiode. Can the authors provide their analysis matched on any reported experimental data for the validation?
  • There are some references not found in the manuscript.

Round 2

Reviewer 1 Report

While the reviewer respectfully acknowledges the explanations and efforts from the authors that did improve the paper to some extent, to keep the standard of this journal regretfully this manuscript still cannot be accepted in its current form.

(1) I suggest that the editors provide some more time to the authors to complete the PL and PLE studies on the coated thin films and add these important data to the revision.  This will greatly improve the quality of the work.

(2) For Question #2, I understand the authors' points. However, since this is a key result/conclusion in the paper, one needs more direct experimental data to support it. The authors could verify this by adding a visible light band-pass filter right before the detector to block the UV light, and compare the transmittance signals with and without perovskite nanoparticles. For the case without the perovskite, the signal should be around noise level, while with perovskite the detector should still see signals from the down-converted photons when measuring "UV" transmittance. This would be much more convincing to present to the readers.

I encourage the authors to complete the experimental work and bring up the level of the paper.

Author Response

We have performed some more experimental work to adress the Reviewer's comments.

(1) I suggest that the editors provide some more time to the authors to complete the PL and PLE studies on the coated thin films and add these important data to the revision.  This will greatly improve the quality of the work.
Given more time by the editors we had a chance to perform the PLE studies of the samples. In the article we have chosen to present the best one which is PSK515 in Zeonex - 120 %. The results are shown in the Figure 18 and commented in the lines 262-272.

(2) For Question #2, I understand the authors' points. However, since this is a key result/conclusion in the paper, one needs more direct experimental data to support it. The authors could verify this by adding a visible light band-pass filter right before the detector to block the UV light, and compare the transmittance signals with and without perovskite nanoparticles. For the case without the perovskite, the signal should be around noise level, while with perovskite the detector should still see signals from the down-converted photons when measuring "UV" transmittance. This would be much more convincing to present to the readers.

We have performed the experiment using the UV filter to block the UV light from reaching the nanoparticles and compared the signal with the measurement when the UV light has not been blocked. A visible difference can be noted whereas for the base layer in which no particles are present, these two measurements give the same results. This is presented in Figure 19 and commented in lines 273-283.

Once again, we are thankful for the Reviewer's comments and research suggestions. It has pushed us forward in the layers investigation. We hope that the paper can be accepted in this form.

Reviewer 2 Report

The authors have addressed my questions and I think the manuscript can be accepted in the current format.

Author Response

Thank you. According to the second Reviewer comments we have also added some PLE measurements of the best layer as well as performed one additional experiment.